# Characterization of Enterobacter phage vB_EcRAM-01, a new *Pseudotevenvirus* against *Enterobacter cloacae,* isolated in an urban river in Panama

Ednner E. Victoria-Blanco[1,2,3ᴑ], Jean Pierre González-Gómez[4ᴑ], Juan Raúl Medina-Sánchez[2], Alexander A. Martínez[1,3,5], Nohelia Castro del Campo[4], Cristóbal Chaidez-Quiroz[4], Jordi Querol-Audi[1,2,3,6]*, Alex Omar Martínez-Torres[2,3,7]*

1 Facultad de Medicina, Programa de Maestría en Ciencias Biomédicas, Universidad de Panamá, Panamá, Panamá, 2 Laboratorio de Microbiología Experimental y Aplicada y Microbiología de Aguas (LAMEXA-LAMA), Universidad de Panamá, Panamá, Panamá, 3 Sistema Nacional de Investigación (SNI), Secretaría Nacional de Ciencia, Tecnología e Innovación (SENACYT), Panamá, Panamá, 4 Laboratorio Nacional para la Investigación en Inocuidad Alimentaria (LANIIA), Centro de Investigación en Alimentación y Desarrollo, A. C. (CIAD), Culiacán, Sinaloa, México, 5 Genomics and Proteomics Research Department, Gorgas Memorial Institute of Health Studies, Panamá, Panamá, 6 Facultad de Medicina, Departamento de Bioquímica y Nutrición, Universidad de Panamá, Panamá, Panamá, 7 Facultad de Ciencias Naturales, Departamento de Microbiología y Parasitología, Exactas y Tecnología, Universidad de Panamá, Panamá, Panamá

ᴑ These authors contributed equally to this work.
* alex.martinez@up.ac.pa (AOMT); jordi.querol@up.ac.pa (JQ-A)

**Data Availability Statement:** The datasets presented in this study can be found in online

## Abstract

The *Enterobacter cloacae* complex, a prominent bacterium responsible worldwide for most bloodstream infections in the hospital environment, has shown broad-spectrum antibiotic resistance, including carbapenems. Therefore, bacteriophages have again attracted the attention of the science and medical community as an alternative to control Multidrug resistant bacteria. In this study, water samples from Río Abajo River, in Panama City, Panama, were collected, for phage isolation, purification, characterization and propagation against the *E. cloacae* complex. As result, a phage produced clear and round plaque-forming units indicating a lytic phage was isolated. Further analyses concluded that this phage is stable at temperatures between 25°C and 50°C, it remains infective in a pH range between 7 to 11, with high sensitivity to Ultraviolet light. Remarkedly, it exhibits a narrow host specificity only infecting *E. cloacae*. Whole genome sequencing revealed that is a myovirus with a genome size of 178,477 bp, a G-C content of 45.8%, and containing approximately 294 genes. Among them, protein-encoding genes involved in morphology, inactivation, adsorption to cells, DNA injection and lytic enzymes were identified. Additionally, the genome contained two tRNA sequences. Genes that encode holins and endolysins, typical of lytic bacteriophages, were also present. A whole-genome sequencing analysis indicated that, according to the genus demarcation criteria, this phage belongs to a novel species within the Family *Straboviridae*, called genus *Pseudotevenvirus*.

repositories. The names of the repository/
repositories and accession number(s) can be
found at https://www.ncbi.nlm.nih.gov/genbank/,
OL551674, SAMN23426491, SRR18344380.

**Funding:** This work was funded in part by the
Sistema Nacional de Investigación (SNI) of the
Secretaría Nacional de Ciencia, Tecnología e
Innovación (SENACYT, Panama), through grants
87-2019, 81-2019 and FID18-044, and also from
the Water, Protected Areas and Wildlife Trust of the
Ministry of Environment of Panama through grant
026-45-2019 and grant 111130406 from the
Ministry of Economy and Finance of Panama. The
funders had no role in study design, data collection
and analysis, decision to publish, or preparation of
the manuscript.

**Competing interests:** The authors have declared
that no competing interests exist.

## Introduction

*Enterobacteriaceae* are ubiquitous in nosocomial settings and can play important roles in the burden of disease in the urinary tract, lower respiratory tract, intra-abdominal organs, and skin infections. The most relevant members of this Gram-negative bacterial group are *Klebsiella pneumoniae*, *Escherichia coli*, and *Enterobacter* spp. The severity of the infections is intensified by the resistance level to most available antibiotics, including carbapenem and cephalosporins [1]. Special attention to *Enterobacter* is due to the intrinsic resistance to ampicillin and broad-spectrum cephalosporin which reduces the alternatives for antibiotic treatments. *Enterobacter* belongs to the ESKAPE group (acronym for *Enterococcus faecium*, *Staphylococcus aureus*, *Klebsiella pneumoniae*, *Acinetobacter baumannii*, *Pseudomonas aeruginosa*, and *Enterobacter* spp), related worldwide to severe nosocomial infections with a highly increasing multi-drug resistance. A problem of epidemiological importance is the presence of AmpC-producing strains together with BLEE-encoding plasmids and colistin co-resistance in *E. cloacae* complex (ECC) strains [2–4]. The continuous emergence of multi-drug resistant strains have hampered the effectiveness of antibiotic treatments. Therefore, one possible approach to address this problem is the use of bacteriophages. In itself, the magnitude of the impact of phage predation on pathogenic bacteria has been demonstrated. In recent years, phage therapy has been successfully applied to *E. coli* O157:H7, causative agent of hemorrhagic colitis and hemolytic uremic syndrome [5–7]. The use of phages has also been studied in the treatment of peritoneal infections, otitis externa, in the eradication of *P. aeruginosa* in patients with cystic fibrosis [8,9], in the treatment of urinary infections or osteomyelitis caused by *S. aureus* [10]. The favorable effects of phage therapy may even exceed its antibacterial activity. A surprising number of data has demonstrated the *in vivo* activity of phages such as the adenovirus growth suppression mediated by bacteriophage T4 [11] or development inhibition of *P. aeruginosa* by phage Pf4 [12].

In recent years, several studies have been published on the use of bacteriophages against *E. cloacae*. Isolation and Characterization of Lytic Phages: A recent study focused on the isolation of the phage ECLFM1 from sewage water using a carbapenem-resistant clinical strain as the host. ECLFM1 exhibited rapid adsorption and a 15-minute latent period, with a burst size of approximately 75 PFU/infected cell. This phage belongs to the Karamvirus family and demonstrated high effectiveness in therapeutic applications, increasing survival rates in zebrafish infected with E. cloacae [13]. Another study identified and characterized the phage vB_EclM_Q7622, isolated from sewage in a poultry farm. This phage showed remarkable capability to infect and destroy multi-drug-resistant strains of *E. cloacae*, highlighting its potential as a therapeutic agent in clinical infections [14]. Phages PφEn-CL and PφEn-HO demonstrated the ability to eliminate multi-drug-resistant *E. cloacae* strains. The study emphasized the specificity of these phages and their efficacy in destroying bacterial biofilms, which is crucial for treating persistent infections in hospital settings [15].

The principal advantage of employing phages in therapy and biological control lies in their specificity towards bacterial targets. Those that do not infect commensal bacteria can be selected, therefore posing no threat to humans and animals. Their widespread presence in nature, their versatility, and their inherent resilience have elevated their status from mere research organisms to potentially viable and indispensable tools in the battle against growing antimicrobial resistance. However, the transfer of resistance and virulence genes, and other genes associated with negative outcomes through bacteriophages, constituted to an entire research domain aimed at mitigating any potential adverse impacts [16,17]. For these reasons, this study is aimed to isolate and biologically and genetically characterize a phage against the *E. cloacae* complex.

## Methods

### Bacterial strains and growth conditions

The host strain ATCC 23355 of *E. cloacae* complex used for the isolation and proliferation of the phage was the same as previously reported [18]. In addition, *S. aureus*, *E. coli*, *E. aerogenes*, *E. cloacae*, and *P. aeruginosa*, provided by the Human Microbiology Department, Faculty of Medicine, Universidad de Panamá; Hospital Santo Tomas, and Complejo Hospitalario Dr. Arnulfo Arias Madrid, were used for host range determination. The *E. cloacae* host strain, along with twenty-seven clinical isolates of *E. cloacae*, were tested for antibiotic susceptibility using the Vitek II Automated Microbiological System (BioMérieux, Marcy l'Étoile, France) (results shown in S1 Table). The breakpoints were applied according to the Clinical and Laboratory Standards Institute (CLSI) guidelines [19].

### Ethics statement

The Bioethics Committee of the Universidad de Panamá with the endorsement of the Dirección General de Salud Pública of the Ministerio de Salud (Ministry of Health), approved this study (Sequential: 617).

### Phage isolation

The Enterobacter phage vB_EcRAM-01 was isolated from waters of Rio Abajo river in Panama City, Panamá (coordinates, 9°00′55.2″N 79°29′25.6″W), using *E. cloacae* strain ATCC 23355 as a propagation host, which was grown previously in tryptic soy broth at 37˚C for 24 h. Phage isolation, propagation, and purification were performed using the soft-agar overlay method [20]. Briefly, phage purification process involved serial dilution of filtrates exhibiting lytic activity against *E. cloacae*. Specifically, 100 μL of the phage dilution was mixed with an overnight culture of the host bacterium in 3 mL of soft agar (TSB with 0.4% agar), poured onto TSA plates, and incubated at 37˚C for 24 h. The phage plates, selected based on size and clarity, were transferred to microtubes containing 1 mL of nanopure water and the whole process was repeated at least three times to confirm that the isolated phages were descendants of a single virion.

### Phage concentration

The soft-agar overlay method was used to propagate the isolated phage. 5 mL of phage buffer (40 mM Tris-Cl, 150 mM NaCl, 10 mM $MgSO_4$) were added to each Petri dish and left for 24 hours at 4˚C. Then the buffer was recovered with a serological pipette and subsequently, the top agar was scraped. Sample was centrifuged at 4,000 rpm for 15 minutes and the supernatant was recovered and centrifugated again was. The supernatant was recovered and filtered using a 0.22 μm filter. The phage sample was then incubated with PEG8000 (8%w/v final concentration) in an automatic tube mixer O/N at 4˚C. Sample was centrifuged at 6,000 rpm for 15–30 minutes, the supernatant was discarded, and the pellet was gently resuspended with 1 mL of sterile PBS buffer.

### Host range of vB_EcRAM-01 against clinical *E. cloacae* isolates: Spot test and efficiency of plating (EOP)

Clinical isolates of *E. cloacae*, *S. aureus*, *E. coli*, *P. aeruginosa*, and *E. aerogenes*, along with *E. cloacae* and *E. aerogenes* ATCC reference strains, were cultured until reaching the exponential growth phase ($OD_{600nm}$ 0.6). Subsequently, 200–500 μL of each culture were mixed with 3 mL

of top agar containing 10 mM $CaCl_2$ and plated onto a 1.5% nutrient agar plate. After solidification, a single droplet (10 μL) from each phage dilution ($10^{-1}$ to $10^{-8}$) was applied onto the agar [21]. The droplets were allowed to dry for 10 min and plates were incubated at 37°C for 24 h—48 h. A phage titer in PFU/mL was calculated for each strain tested [22]. Phage virulence was assessed using the efficiency of the plating (EOP) method. The EOP was determined by dividing the phage titer at the final dilution on the test strain by the phage titer on its original isolation strain. Based on these calculations, phages were categorized into highly virulent ($0.1 < EOP > 1.00$), avirulent but still active ($EOP < 0.001$), or completely avirulent (no plaques were observed).

## Morphological characterization

The isolated phage underwent examination using transmission electron microscopy (TEM). A formvar-coated grid (400 mesh copper grid) was utilized, onto which a drop of $10^9$ PFU/mL phage stock was applied. Subsequently, negative staining was performed with 2% phosphotungstic acid (pH 7.2) for 5 minutes, and excess stain was eliminated with filter paper. Electron micrographs were captured at varying magnifications using a JEOL JEM-1011 transmission electron microscope (20).

## Sodium dodecyl sulphate polyacrylamide gel electrophoresis (SDS-PAGE)

SDS-PAGE electrophoresis was performed following the Laemmli method [23]. Separation was carried out in 10% resolving gel in standard Tris–glycine buffer at a constant current of 20 mA. A molecular weight standard (Precision Plus protein ladder, BioRad) was used in the electrophoretic separation, after which the gels were stained with Coomassie blue (Sigma, Ge). After incubation with a solution containing 20% ethanol and 7% acetic acid to remove the excess of stain, images were recorded using a BioDoc-it Imaging System (UVP).

## Effects of UV light, temperature and pH on the viability of phage vB_EcRAM-01

Purified vB_EcRAM-01 was exposed to different physicochemical conditions (UV light, temperature and pH) Suspensions were exposed to ultraviolet light at a distance of 50 cm for 10, 20, 40, 60, 70, and 90 s. To determine the effect of temperature, phage suspensions were placed at 25°C, 37°C, 50°C and 70°C, respectively. pH stability was determined by mixing assay vB_EcRAM-01 phage samples resuspended in SM-gelatine buffer (50 mM Tris–HCl pH 7.5, 100 mM NaCl, 8 mM MgSO4 and gelatin 2% (w/v)) with a pH ranging from 2 to 14. Phage viability was determined by counting the number of lysis PFU using the double-layer agar technique and the experiments were carried out in triplicates [24,25].

## Bacterial reduction assay

Bacteriolytic activity was evaluated using a spectrophotometer, targeting *E. cloacae* ATCC 23355. The bacterial strain's overnight culture, adjusted to 1.0 $OD_{600nm}$, served as the inoculum. A concentration of 8 log PFU/mL was tested with appropriate phage dilutions. Each tube, including negative and positive controls, contained 20 mL of TSB. Experimental tubes received an additional 200 μL of the phage dilution, followed by the addition of 1000 μL of bacterial culture to each tube (excluding the negative control). Incubation was carried out for 180 minutes at 37°C, with measurements at 600 nm taken every 15 minutes. The experiment was conducted in triplicates.

## Bacteriophage DNA isolation and genome sequencing

The QIAamp MinElute Virus Spin kit (QIAGEN, Valencia, CA) was used to extract DNA, according to the manufacturer´s instructions. The purity was assessed spectrophotometrically at 260 and 280 nm, using a BioSpec-nano Micro-volume UV-Vis Spectrophotometer (Shimadzu, Kyoto, Japan).

The DNA libraries were prepared using the Nextera XT DNA Library Preparation Kit (Illumina, San Diego, CA, USA) through fragmentation, PCR amplification, PCR cleanup, and library normalization. The genome was sequenced at the Genomics and Proteomics Research Department of ICGES using the MiSeq Illumina sequencer (2 x 250 bp paired-end reads) (ThermoFisher Scientific Waltham, Massachusetts USA). Raw reads (629,372 reads in total) were checked for quality using FastQC (www.bioinformatics.Babraham.ac.UK/projects/fastqc) and trimmed using Timmomatic v0.39 [26].

Genome annotations, comparative genomics and phylogenetic analyses

The genome was assembled through SPAdes v3.15.1 [27], resulting in a single contig with 8.96-fold coverage [27]. We mapped the reads against the contig obtained using Bowtie v0.9.6 (http://bowtie-bio.sourceforge.net/index.shtml) to corroborate the sequence ends [28]. The phage genome annotation was performed using RASTk v2.0 [29–31]. Gene calling was performed through GeneMarkS v4.28 (http://exon.gatech.edu/GeneMark/genemarks.cgi) [32], and Glimmer v3.02 (http://ccb.jhu.edu/software/glimmer/index.shtml) [33,34].

Several methods were used to annotate the ORF functions, including the protein basic local alignment search tool (blastp) on the NCBI server (https://blast.ncbi.nlm.nih.gov/Blast.cgi) based on the search of the non-redundant protein sequence database [35], Hhpred using structural/domain database (https://toolkit.tuebingen.mpg.de/tools/hhpred) [36–39] and HMMER v2.41.1 through HMM database [40,41]. (https://toolkit.tuebingen.mpg.de/tools/hhpred). Using a non-redundant protein matrix, employing Markov hidden model profiles [42], the annotations were manually curated using the Geneious v9.1.8 platform employing default parameters. Identification of transfer RNAs was performed using tRNA scan-SE v2.07 (http://lowelab.ucsc.edu/tRNAscan-SE/) and ARAGORN v1.2.41 [43] software (http://www.ansikte.se/ARAGORN/). Comparative analysis between genomes was performed using Mauve progressive alignment v2.3.1 [44] (http://darlinglab.org/mauve/mauve.html) employing default parameters.

Pairwise comparison of nucleotide sequences was performed using the Genome-BLAST Distance Phylogeny (GBDP) method of the VICTOR (Virus Classification and Tree Building Online Resource) platform [45] (https://ggdc.dsmz.de/victor.php), in the recommended environments for prokaryotic viruses. The resulting intergenomic distances were used to infer a branch-supported balanced minimum evolution tree using FASTME, including SPR post-processing [46] using D6 formula. Branch support was inferred from 100 pseudo-bootstrap replicates each. Trees were rooted at the midpoint [47], visualized using FigTree; personalized with taxon boundaries at species, genus, and family scales; and estimated with the iTOL program [48] (https://itol.embl.de/). The recommended clustering threshold is thirthy nine [45] and an F-value (fraction of links required for cluster fusion) of 0.5 [45].

Determination of taxonomy and viral replication cycle type was achieved using PATRIC v3.6.9 [49] (https://patricbrc.org/) and PhageAI (https://phage.ai/) server [50]. The genome comparison between Enterobacter phage vB_EcRAM-01 and close relatives was performed using EasyFig v2.2.2 [51]. At the end, the intergenomic similarities amongst complete viral genomes were calculated through the VIRIDIC web tool [52] with blastn default settings.

## Statistical analysis

Statistically significant differences were determined using ANOVA. Differences in group means were compared using Tukey's HSD test. A P-value below 0.05 was considered statistically significant. Statistical analysis was conducted using R version 4.3.1 [53].

# Results

## Bacteriophage isolation, host range analysis and morphology characterization

In recent years, several studies have been published on the use of bacteriophages against E. *cloacae*, an opportunistic pathogen posing a clinical challenge due to its resistance to multiple antibiotics [13–15]. In this study, the Enterobacter phage vB_EcRAM-01 was isolated from the waters of the Rio Abajo river in Panama City, Panama. Using *E. cloacae* strain ATCC 23355 as the host, the phage was isolated and purified. The isolation involved a series of dilutions and the use of the soft-agar overlay technique to propagate the phage. Clear plaques indicating phage lytic activity were selected and purified to ensure that the isolated phages originated from a single virion.

In order to evaluate the host specificity of vB_EcRAM-01, a comprehensive assessment was conducted involving a total of thirty-three strains: two ATCC strains alongside a diverse selection of, both, Gram-negative and Gram-positive clinical isolates (Fig 1).

Notably, vB_EcRAM-01 exhibited the capacity to infect all the *E. cloacae* isolates with varying degrees of efficacy while displaying negligible antimicrobial activity (efficiency $< 0.10$) against the other species tested, which included *E. coli*, *E. aerogenes*, *A. baumannii*, *S. aureus*, and *P. aeruginosa*. Upon infecting the reference strain, the clear plaque-forming units (PFUs), characterized by smooth perimeters, measured within the range of 3 mm to 7 mm in diameter, a hallmark indicative of its lytic phage attributes (S1 Fig). TEM micrographs of the phage showed a morphology typical of a myovirus, a phage with a tail, belonging to the class *Caudoviricetes*. The size measurements of its icosahedral head are about 104 x 76 nm, and its contractile tail is approximately 110 nm (S2 Fig). Previously, phage morphology was sufficient to classify them taxonomically, but nowadays bioinformatics tools are used to perform similarity analysis and group them by evolutionary distances.

## Environmental stress resistance: Stability against UV light, temperature and pH

Exposure to UV light (365 nm) decreased the infectivity of phage vB_ECRAM-01 over time. Results showed an exponential decay resulting in complete inactivation when incubated for more than one minute (Fig 2A).

The phage exhibited lytic activity when subjected to temperatures ranging from 25°C to 50°C, even after being incubated for 90 minutes (Fig 2B). However, when exposed to 70°C, the phage became completely inactivated after only 20 min of incubation and, at 80°C, it lost its infectivity entirely (data not shown for clarity).

The stability of vB_EcRAM-01 was determined at various pH values. It maintained its infectivity in a pH range between pH 7.5 and pH 11. The best conditions were found between pH 9 and 10 (Fig 2C). Incubation under acidic pH resulted in a significant reduction of viral proliferation. The phage was not able to replicate in a pH greater than 11 or less than 3. Overall, the best activity was found in alkaline media.

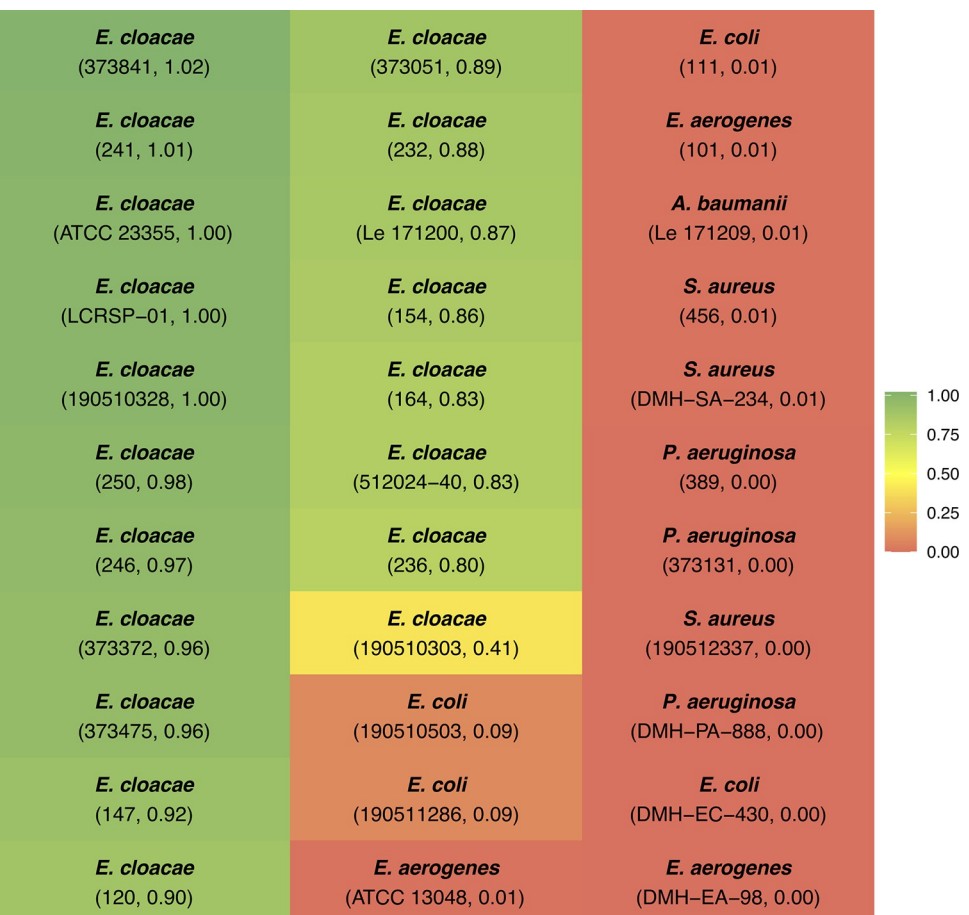

**Fig 1. Host range specificity and efficiency of plating.** The strain code followed by the efficiency value are shown in parentheses. Color range code associated with EOP is indicated on the right.

## Bacteriolytic activity

The evaluated phage demonstrated a robust, complete bacteriolytic effect in the initial 105 minutes of the assay. Following this, the OD curve started to rise due to an increase in bacterial cell numbers. This trend persisted in subsequent measurements, although the rate of increase was lower than that observed in the positive control (Fig 3).

## Comparative genomics and phylogenetic analysis

The genomic attributes of phage vB_EcRAM-01 were delineated as follows: the genome spans 178,477 base pairs with a 45.0% of GC content, hosting a total of 293 ORFs and two tRNA. Among these ORFs, 86 genes were implicated in anticipated functions encompassing morphology, inactivation, cellular adsorption, DNA injection, and host lysis (Fig 4, S1 Table).

Through a UniProt database blast search, we were able to identify to structural proteins, one sharing 97% homology with the Major capsid protein (ID: A0A1B1IX95) of Citrobacter phage vB_CfrM_CfP1, having a mass of 56.37 kDa [54] while the other protein exhibits 96.7% homology with the Baseplate wedge subunit protein (ID: M9V1F2) from Escherichia phage Lw1, with a mass of 38.01 kDa [55]. In fact, analysis by polyacrylamide gel electrophoresis revealed two prominent protein bands with molecular masses close to 50 and 37 kDa,

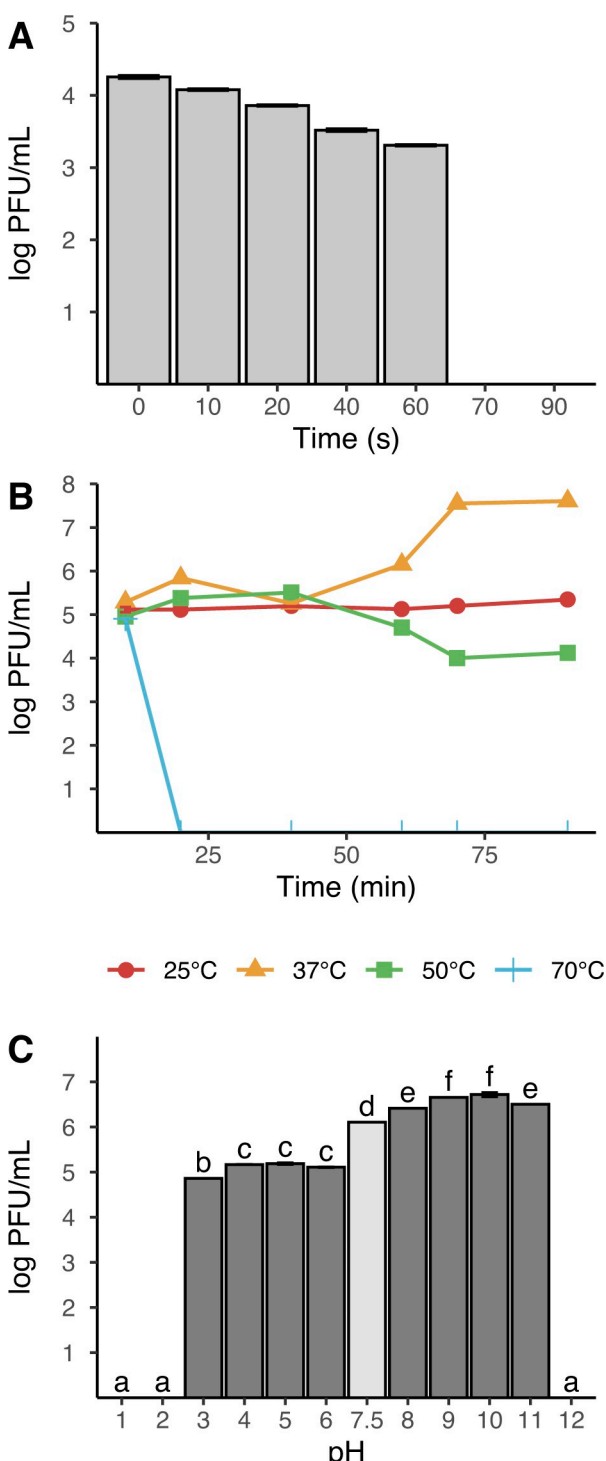

**Fig 2.** Stability of vB_ECRAM-01 to (A) UV radiation, (B) temperature, and (C) pH. Data is expressed as mean ± standard error. Treatments with the same letters in panel (C) are not significantly different (Tukey's HSD test, p>0.05).

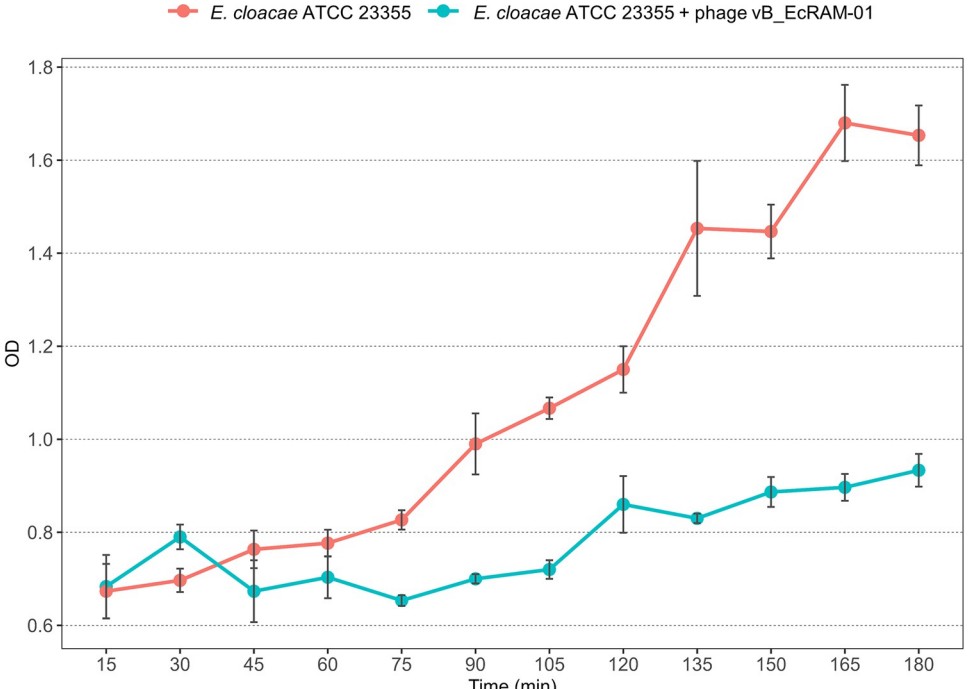

**Fig 3. Effect of vB_EcRAM-01 on the growth of the *E. cloacae* complex.** The light blue line indicates the bacterial concentration without being infected with the virus at 37°C, showing an absorbance of 1.6. On the other hand, the orange line shows the concentration of the bacteria being infected from the beginning, with a reduced absorbance of up to 0.77 (ANOVA F = 613.62. g.l = 1 and p<0.0001).

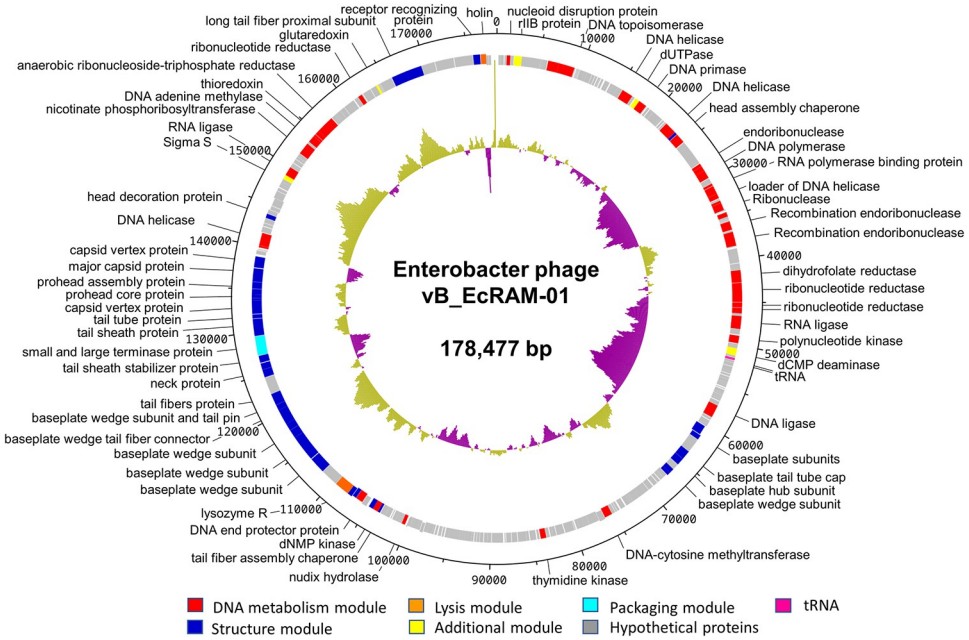

**Fig 4. Genomic organization of phage vB_EcRAM-01.** The inner circle represents GC content. Arrows indicate annotated ORFs color-coded according to their putative function.

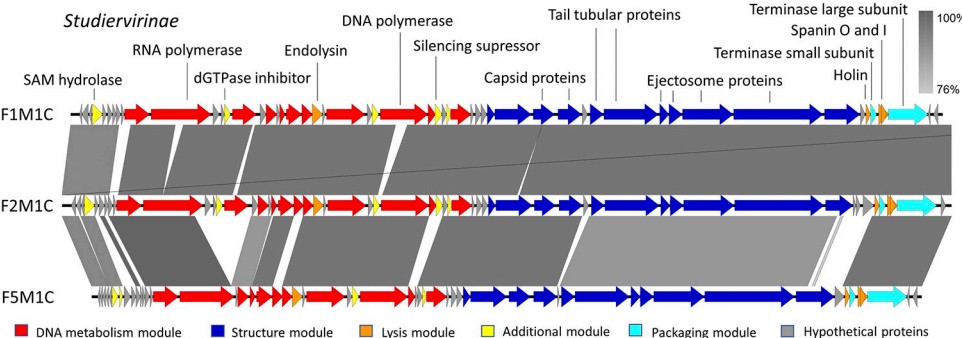

**Fig 5. Comparison of Enterobacter phage vB_EcRAM-01 genome with close relatives.** Color scales represent the percentage of nucleotide identity between regions obtained by blastn.

respectively (S3 Fig). In bacteriophages possessing both capsid and tail structures, SDS-PAGE typically reveals multiple proteins, including capsid, tail, and baseplate proteins. Capsid proteins form the outer shell encasing the phage genome, while baseplate proteins are situated at the tail base, facilitating recognition and binding to specific receptors on host cell surfaces [56]. Hence, it is plausible that the observed proteins serve these functions; however, precise identification through LC-MS/MS analysis would provide a more definitive comparison [57].

Remarkably, the genome of vB_EcRAM-01 was lacking discernible genes associated with resistance, virulence, or allergenicity. The genomic blueprint encodes two transfer RNAs (tRNAs), one corresponding to methionine (cat) and the other to glycine (tcc). A pertinent insight is the substantial sequence identity—90.11%—shared with other phages (Fig 5) [18].

To assign a taxonomic classification to phage vB_EcRAM-01, a phylogenetic tree based on whole-genome was constructed, and intergenomic similarities were calculated between the closely related genomes reported in GenBank. The phylogenetic tree and intergenomic similarity heatmap revealed that phage vB_EcRAM-01 is a new member of the *Pseudotevenvirus* genus within the *Straboviridae* family.

The GBDP phylogenetic tree using the D6 formula with an average support of 31%, identifying ten and nine clusters, respectively, at the genus level. At the family level, there were five and six clusters, respectively (Fig 6).

Nucleotide identity analysis (ANI) was done on the 28 genomes with the highest homology in the GenBank database using VIRIDIC. According to the results of the heat mapping, phage vB_EcRAM-01 shares less than 95% nucleotide identity with previously reported genomes, thereby establishing it as a novel species within the Family *Straboviridae*, genus *Pseudotevenvirus*, according to the genus demarcation criteria (>70% nucleotide identity over the entire genome length) recommended by the ICTV (Fig 7).

Comparative BLASTn analysis of the complete genome of vB_EcRAM-01 against phages in the GenBank database revealed that vB_EcRAM-01 has a high degree of coverage and identity with the Enterobacter phages ENC22 (OL355128.1), ENC7 (OL355125.1), ENC25 (OL355127.1), EBPL (MT341500.1). In VIRIDIC analysis, the phages shared collinear gene blocks that are identical in length between the two genomes. In comparison, other blocks revealed a high degree of similarity between this phage and others, including Cronobacter phage vB_CsaM IeE (NC_048646.1), Cronobacter phage vB_CsaM IeN (KX431560.1), Citrobacter phage Margaery (KT381880.1), and myoviral members of the subfamily *Tevenvirinae*.

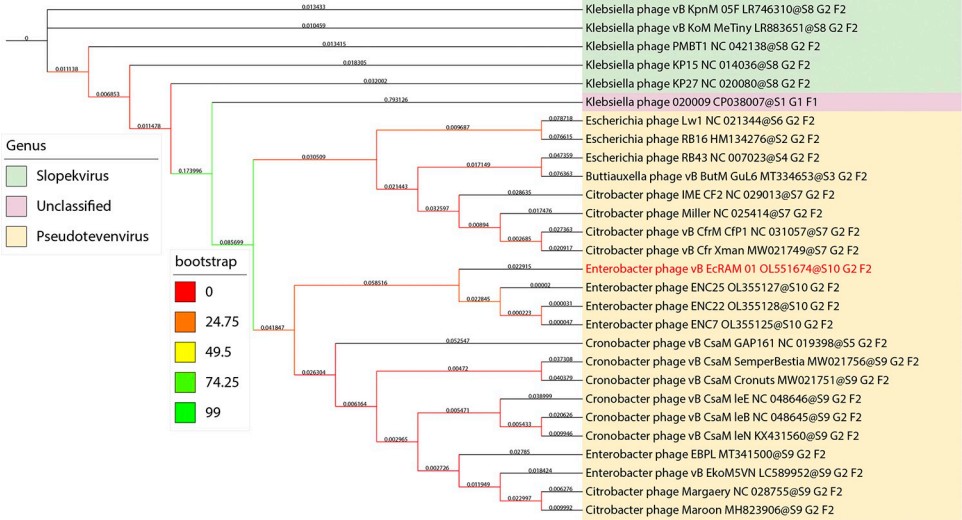

**Fig 6. Phage vB_EcRAM-01 genomic and phylogenetic analyses.** Genome Blast Distance Phylogeny (GBDP)-based phylogenetic tree of vB_EcRAM-01 and 28 related Straboviridae members created by iTOL v6.0 platform. The Enterobacter phage vB_EcRAM-01 belongs to the genus Pseudotevenvirus and is a new phage species.

## Discussion

In Panama, seven out of the eight bacterial pathogens responsible for 80% of the reported infections concerning the emergence of resistance in a hospital level (ESKAPE) has been identified: *S. aureus* (16%), *Enterococcus* spp (14.4%), *E. coli* (12%), coagulase-negative

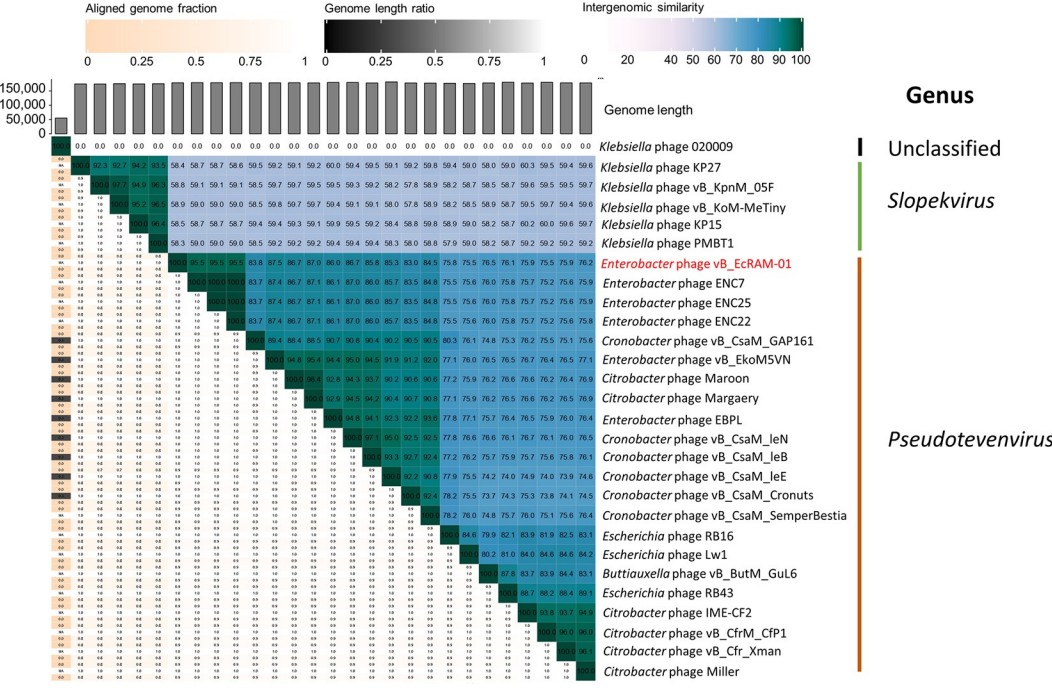

**Fig 7. Taxonomic classification of Enterobacter phage vB_EcRAM-01.** A heatmap created by VIRIDIC based on intergenomic similarities between viral genomes, with the proportion of similarity shown by the color scale.

*Staphylococcus* (11%), *K. pneumoniae* and *K. oxytoca* (8%), *P. aeruginosa* (8%), and *Enterobacter* spp. (5%). Several microorganisms developed antibiotic resistance to the more common antibiotics to treat these bacterial infections like piperacillin-tazobactam, cephalosporins, nitrofurans, and carbapenems [58,59].

Therefore, in comparison to the usage of antibiotics, phage therapy is a viable option. While scientific publications on phage isolation have increased in recent years, only a small number of phages have been commercialized [60]. One of the primary reasons is a lack of adequate characterization, which is required to detect and mitigate the hazards associated with phage therapy [60,61]. Moreover, the biological characteristics of a phage determines its stability and therefore, its optimum administration forms [61].

As a first step, vB_EcRAM-01was subjected to environmental stress tests, such as temperature, UV radiation and pH, factors that play a role in phage survival [62,63]. The results indicate that vB_EcRAM-01 is stable at temperatures ranging from 25°C to 50°C, although its replication rate is highest at 37°C. This aligns with findings observed in Myoviridae phages [64,65] exhibiting a comparable stability profile within a broad temperature range. However, it deviates from the EBP phage, known for its resilience across a wider temperature span [66]. Notably, inactivation under UV light transpires rapidly, underscoring the necessity of safeguarding possible clinical applications from sunlight exposure. Interestingly, vB_EcRAM-01 maintains its infectivity across expansive pH and temperature ranges, implying potential robustness within the human body. Enhancements through administration systems designed to shield the phage from gastrointestinal pH could potentially further increase its utility and stability in therapeutic applications [67].

The primary objective of this study was to investigate the host specificity (host range) of phage vB_EcRAM-01 concerning isolates within the *E. cloacae* complex. Accurate identification of the host bacteria's specificity for bacteriophage vB_EcRAM-01 is crucial for a comprehensive characterization of this bacteriophage in order to evaluate a potential therapeutic use. Our observations indicate that this bacteriophage displayed reactivity only with its host strain. These results are in line with findings reported for phages TSE1, TSE2, and TSE3 [68]. *E. coli* displayed some sensitivity but did not exhibit complete lysis. Exceptions to this pattern were observed with myophages *Enterobacter* phage ENC22, *Enterobacter* phage ENC7, and *Enterobacter* phage ENC25, which demonstrated the ability to infect diverse *Enterobacter* species [69].

Apart from meeting the requirements of being innocuous to normal microbiota, therapeutic use involves the employment of fully lytic phages that lack undesired virulence factor genes or antibiotic resistance genes [70]. The genomic assessment of the phage vB_EcRAM-01 confirmed its status as a strictly lytic phage according to the PhageAI platform. The genome contained genes coding for holins and endolysins, offering several advantages. These enzymes collaborate to facilitate bacterial lysis, enabling the release of newly assembled viral particles and the termination of the bacterial host [71]. Additionally, holin proteins play a crucial role in regulating the timing of lysis, ensuring that it occurs at the appropriate stage of the phage life cycle [72]. An important aspect to note is that resistance to endolysins has not been reported to date [73]. This suggests a higher probability of success when employing phages containing endolysins. Besides, it lacked genes associated with lysogeny, such as integrases, transposons, and excitons [74].

The evolutionary context designated it as a T4-type myophage spanning a genome length of 178,477 base pairs. Computational analysis further classified it as a viral entity within the Class *Caudoviricetes*, *Strabovviridae* family, and *Pseudotevenvirus* genus [74]. This type of virus is characterized by the presence of tRNA. Due to the compact nature of phage genomes, which constrains their capacity to utilize host resources fully, these genetic elements may serve as

compensation for compositional differences between phages and hosts. They have the potential to supplement the translation machinery with their genetic information, thereby improving fitness and fortifying virulence [75].

By scanning the GenBank database for homologous sequences, a 92% sequence homology with myophage EBPL was observed, a phage that also infects *E. cloacae* [66], and 99.16% sequence homology with myophages ENC7, ENC25, and ENC22 [69].

According to genotypic characteristics [76], vB_EcRAM-01, like an EBP phage, is a myovirus-like phage [66]. The vB_EcRAM-01 virus was termed a bacterial infecting virus isolated from Rio Abajo waters that suppressed *E. cloacae*, a myophage, according to the informal phage nomenclature criteria [77–79].

In this study, the genome-wide syntenic analysis indicated that both phages had comparable rearrangements but differed slightly in nucleotide sequence. Additionally, a phylogenetic study of whole-genome sequences showed that both phages diverged from a common ancestor but experienced many nucleotides substitution events since then [80]. Given that EBPL was previously described as a singleton phage in the myophage phylogenetic tree [81], phylogenetic and nucleotide identity analyses indicated that vB_EcRAM-01 might represent a previously unreported species.

In Conclusion, this study identified a lytic bacteriophage. The characterization enhances comprehension of this novel phage, and through its phylogenetic relatives, it may exhibit a myovirus-like morphology, as demonstrated by insights from next-generation sequencing. This sequencing elucidates that vB_EcRAM-01 aligns with the Myoviridae family in accordance with the conventional taxonomic classification [82]. According to the phylogenetic analysis previously described, this virus is a new member of the family *Straboviridae* and the genus *Pseudotevenvirus* according to the new classification [83]. The vB_EcRAM-01 strain is also a viable candidate for many clinical applications, including phage therapy and biologic control or decontamination, due to its host specificity, the genetic link, and lack of lysogenic genes.

## Supporting information

**S1 Fig. Lytic effect of bacteriophage vB_EcRAM-01.** According to the double agar plate method, the bacterial lawn of the *E. cloacae* complex is infected by bacteriophages. After purification, lysis PFU measuring approximately 9 mm (shown in the figure), rough-edged and completely clear, were obtained.
(TIF)

**S2 Fig. Overall morphology of vB_EcRAM-01.** A representative micrograph of a purified sample of the bacteriophage is shown (scale bar: 100nm).
(TIF)

**S3 Fig. Electrophoretic profiles (SDS-PAGE) of phage EcRAM-01.** M: Precision Plus protein ladder., S: Phage sample. Ladder molecular weights are indicated on the left.
(TIF)

**S1 Table. Annotation of Enterobacter phage EcRAM-01 genome.**
(DOCX)

## Acknowledgments

Host bacterial strains were provided by José E. Moreno, MSc., of the Microbiology Division of Laboratorio Central de Referencia en Salud Pública, Instituto Conmemorativo Gorgas de Estudios de Salud (ICGES). We extend our gratitude to the staff of the Microbiology Department

at the Clinical Laboratory of Hospital Santo Tomas for providing us with the clinical bacteria strains.

Additionally, we would like to thank Claudia González, MSc, Jessica Gondola, MSc, Ambar Moreno, MSc, and Oris Chavarria, MSc, of the Research Department of Genomics and Proteomics of ICGES for providing us with NGS Illumina sequencing. Furthermore, we would like to express our gratitude to Leandra Gómez, MSc, of the Department of Pharmacology at the Faculty of Medicine, Universidad de Panamá, for her invaluable contribution in reviewing the bioethical considerations within this research. Also, thank you to Bs. Tatiana A. Martínez González for her contribution to the English revision of the present work.

## Author Contributions

**Conceptualization:** Alex Omar Martínez-Torres.

**Data curation:** Ednner E. Victoria-Blanco, Jean Pierre González-Gómez, Juan Raúl Medina-Sánchez, Alexander A. Martínez, Jordi Querol-Audi.

**Formal analysis:** Jean Pierre González-Gómez, Nohelia Castro del Campo, Cristóbal Chaidez-Quiroz, Jordi Querol-Audi, Alex Omar Martínez-Torres.

**Funding acquisition:** Jordi Querol-Audi, Alex Omar Martínez-Torres.

**Investigation:** Ednner E. Victoria-Blanco.

**Methodology:** Ednner E. Victoria-Blanco.

**Project administration:** Alex Omar Martínez-Torres.

**Resources:** Alexander A. Martínez, Cristóbal Chaidez-Quiroz, Jordi Querol-Audi, Alex Omar Martínez-Torres.

**Software:** Jean Pierre González-Gómez, Juan Raúl Medina-Sánchez.

**Supervision:** Jordi Querol-Audi, Alex Omar Martínez-Torres.

**Validation:** Jean Pierre González-Gómez, Juan Raúl Medina-Sánchez, Nohelia Castro del Campo, Cristóbal Chaidez-Quiroz, Jordi Querol-Audi.

**Visualization:** Ednner E. Victoria-Blanco.

**Writing – original draft:** Juan Raúl Medina-Sánchez, Nohelia Castro del Campo, Cristóbal Chaidez-Quiroz, Jordi Querol-Audi, Alex Omar Martínez-Torres.

**Writing – review & editing:** Nohelia Castro del Campo, Cristóbal Chaidez-Quiroz, Jordi Querol-Audi.

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
