## [Decision Letter · Decision Letter 0]

30 Jan 2024

PONE-D-23-39735Characterization of Enterobacter phage vB_EcRAM-01, a new Pseudotevenvirus against Enterobacter cloacae, isolated in an urban river in Panama.PLOS ONE

Dear Dr. Querol-Audi,

Thank you for submitting your manuscript to PLOS ONE. After careful consideration, we feel that it has merit but does not fully meet PLOS ONE’s publication criteria as it currently stands. Therefore, we invite you to submit a revised version of the manuscript that addresses the points raised during the review process.

 Please submit your revised manuscript by Mar 15 2024 11:59PM. If you will need more time than this to complete your revisions, please reply to this message or contact the journal office at plosone@plos.org. Please include the following items when submitting your revised manuscript:A rebuttal letter that responds to each point raised by the academic editor and reviewer(s). You should upload this letter as a separate file labeled 'Response to Reviewers'.A marked-up copy of your manuscript that highlights changes made to the original version. You should upload this as a separate file labeled 'Revised Manuscript with Track Changes'.An unmarked version of your revised paper without tracked changes. You should upload this as a separate file labeled 'Manuscript'.

We look forward to receiving your revised manuscript.

Kind regards,

Alexandra Lianou, M.Sc., Ph.D.

Academic Editor

PLOS ONE

Journal Requirements:

"This work was funded in part by the Sistema Nacional de Investigación (SNI) of the Secretaría Nacional de Ciencia, Tecnología e Innovación (SENACYT, Panama), through grants 87-2019, 81-2019 and FID18-044, and also from the Water, Protected Areas and Wildlife Trust of the Ministry of Environment of Panama through grant 026-45-2019 and grant 111130406 from the Ministry of Economy and Finance of Panama."

"This work was funded in part by the Sistema Nacional de Investigación (SNI) of the Secretaría 

 Nacional de Ciencia, Tecnología e Innovación (SENACYT, Panama), through grants 87-2019, 81-

 2019 and FID18-044, and also from the Water, Protected Areas and Wildlife Trust of the Ministry of 

 Environment of Panama through grant 026-45-2019 and grant 111130406 from the Ministry of 

 Economy and Finance of Panama. Clinical bacterial strains were provided by José E. Moreno, MSc., 

 of the Microbiology Division of Laboratorio Central de Referencia en Salud Pública, Instituto 

 Conmemorativo Gorgas de Estudios de Salud (ICGES) and the Department of Microbiology, Hospital 

 Santo Tomás."

"This work was funded in part by the Sistema Nacional de Investigación (SNI) of the Secretaría Nacional de Ciencia, Tecnología e Innovación (SENACYT, Panama), through grants 87-2019, 81-2019 and FID18-044, and also from the Water, Protected Areas and Wildlife Trust of the Ministry of Environment of Panama through grant 026-45-2019 and grant 111130406 from the Ministry of Economy and Finance of Panama."

5. Please amend the manuscript submission data (via Edit Submission) to include author Alex Omar Martínez-Torres, Cristóbal Chaidez-Quiróz, Nohelia Castro del Campo, Alexander A. Martínez, Juan Raúl Medina-Sánchez, Jean Pierre González-Gómez and Ednner E. Victoria-Blanco.

Additional Editor Comments:

**ACADEMIC EDITOR:**According to the evaluation of two referees and the Academic Editor's opinion, the manuscript reports on the findings of an overall well-designed and -executed study, which contribute to the better understanding of the biology of phages and their utilization as biocontrol agents against pathogens with well-established multi-drug resistance, such as Enterobacter cloacae. Nonetheless, the manuscript should be carefully revised both for format and scientific accuracy, whereas one of the reviewers has also commented on the necessity of conducting some additional experiments. The authors are kindly requested to take into account the suggestions and comments of the reviewers, and proceed with meticulous revision the manuscript, providing a point-by-point response to each one of the raised comments.

Reviewers' comments:

Reviewer's Responses to Questions

**Comments to the Author**

1. Is the manuscript technically sound, and do the data support the conclusions?

Reviewer #1: Yes

Reviewer #2: Yes

2. Has the statistical analysis been performed appropriately and rigorously? 

Reviewer #1: Yes

Reviewer #2: Yes

3. Have the authors made all data underlying the findings in their manuscript fully available?

Reviewer #1: Yes

Reviewer #2: Yes

4. Is the manuscript presented in an intelligible fashion and written in standard English?

Reviewer #1: Yes

Reviewer #2: No

5. Review Comments to the Author

Reviewer #1: The paper offers a structured framework for characterization of Enterobacter phage vB_EcRAM-01, a new Pseudotevenvirus against Enterobacter cloacae, isolated in an urban river in Panama.

The authors have conducted many right experiments, but other experiments must be carried out to complete characterization of isolated phage and show the importance of this work. To help the authors improve their manuscript, we have included both general and detailed comments below:

1. It is necessary to describe phage vB_EcRAM-01 infection cycle.

2. Morphological characterization of phage using transmission electron microscopy (TEM) should be done.

3. Bacterial reduction assay should be carried out.

4. How did you perform purification of phage particles?

5. Have you identified the structural proteins of isolated phage by SDS-PAGE gel electrophoresis? It will be great to have the proteomic analysis of this phage.

6. Please state the GenBank accession no. of phage genome.

7. What is the percentage of a G + C content of phage genomic DNA.

8. Please describe distinguished genes found in the genome of phage vB_EcRAM-01.

9. Correct the list of references. Some citations are not present in the reference list.

Reviewer #2: This manuscript provides data for the biological characterization and genomic analysis of Enterobacter cloacae phage vB_EcRAM-01. Results presented in this study contribute to a greater understanding of the biology of phages and expand our knowledge of Enterobacter cloacae phages. However, the overall format of the article is not consistent with the requirements of the official website of PLOS ONE, please download the manuscript formatting guidelines from the official website of PLOS ONE and modify the article format according to the guidelines.

Major comments

1. Line 19- Enterobacter cloacae should not be abbreviated when it first appears in the keyword section.

2. Line 39- The font of Enterobacteriaceae should be italic, please check the full text.

3. Line 108-110- The experimental conditions in the text are 25 ℃, 50 ℃, 70 ℃ and 90 ℃, but Figure 2B shows 25 ℃, 37 ℃, 50 ℃ and 70 ℃. The temperature unit is not written in Figure 2B. The range of pH in Figure 2C is also inconsistent with the text.

4. Figure 2- The unit of the horizontal coordinate of Figure 2 B should be Time (min).

5. Figure 2- In Figure 2 B, why does the number of phage vB_EcRAM-01 gradually increase at 37 °C?

6. Line 180- Please write down the correct unit of temperature.

7. Line 187-215, 257-266- The description of the genome of phage vB_EcRAM-01 is relatively simple. Please provide more details about the genome.

8. Figure 3- Please add colors to phage EBPL and phage Margaery to better compare phage vB_EcRAM-01 with these two phages.

9. Figure 4, Figure 5- Please change the font of the genus names in italics on Figures 4 and 5.

6. PLOS authors have the option to publish the peer review history of their article (what does this mean?). If published, this will include your full peer review and any attached files.

Reviewer #1: No

Reviewer #2: No

---

## [Author Response · Author response to Decision Letter 0]

25 Mar 2024

Panama, Panama, March 23, 2024

Response to the Editor and reviewers:

• Style requirements have been carefully reviewed.

• Grant numbers have been corrected (see below).

"This work was funded in part by the Sistema Nacional de Investigación (SNI) of the Secretaría Nacional de Ciencia, Tecnología e Innovación (SENACYT, Panama), through grants 87-2019, 81-2019 and FID18-044, and also from the Water, Protected Areas and Wildlife Trust of the Ministry of Environment of Panama through grant 026-45-2019 and grant 111130406 from the Ministry of Economy and Finance of Panama."

• The role of funders has been included 

"This work was funded in part by the Sistema Nacional de Investigación (SNI) of the Secretaría Nacional de Ciencia, Tecnología e Innovación (SENACYT, Panama), through grants 87-2019, 81-2019 and FID18-044, and also from the Water, Protected Areas and Wildlife Trust of the Ministry of Environment of Panama through grant 026-45-2019 and grant 111130406 from the Ministry of Economy and Finance of Panama. Clinical bacterial strains were provided by José E. Moreno, MSc., of the Microbiology Division of Laboratorio Central de Referencia en Salud Pública, Instituto Conmemorativo Gorgas de Estudios de Salud (ICGES) and the Department of Microbiology, Hospital Santo Tomás."

"This work was funded in part by the Sistema Nacional de Investigación (SNI) of the Secretaría Nacional de Ciencia, Tecnología e Innovación (SENACYT, Panama), through grants 87-2019, 81-2019 and FID18-044, and also from the Water, Protected Areas and Wildlife Trust of the Ministry of Environment of Panama through grant 026-45-2019 and grant 111130406 from the Ministry of Economy and Finance of Panama."

• This has been included in the cover letter and removed from the acknowledgments section.

5. Please amend the manuscript submission data (via Edit Submission) to include author Alex Omar Martínez-Torres, Cristóbal Chaidez-Quiróz, Nohelia Castro del Campo, Alexander A. Martínez, Juan Raúl Medina-Sánchez, Jean Pierre González-Gómez and Ednner E. Victoria-Blanco.

• Author’s information has been included.

• Figure captions for Supporting Information have been included at the end of the revised version.

Reviewer#1

The paper offers a structured framework for characterization of Enterobacter phage vB_EcRAM-01, a new Pseudotevenvirus against Enterobacter cloacae, isolated in an urban river in Panama.

The authors have conducted many right experiments, but other experiments must be carried out to complete characterization of isolated phage and show the importance of this work. To help the authors improve their manuscript, we have included both general and detailed comments below:

1. It is necessary to describe phage vB_EcRAM-01 infection cycle.

• We respectfully acknowledge the reviewer's comment regarding the description of the infection cycle of phage vB_EcRAM-01. While we appreciate the importance of understanding the infection dynamics of bacteriophages, we opted not to include a detailed description of the infection cycle using methodologies such as the one-step curve. This decision was based on several factors: Firstly, the high variability in results due to technical errors can significantly impact the accuracy of burst size measurements. Secondly, these techniques require a substantial amount of material, which may not be feasible in all research settings. Instead, we focused our characterization efforts on essential tests such as host range determination, bacteriolytic activity assays, and genomic sequencing, which collectively provide valuable insights into the potential therapeutic applications of phage vB_EcRAM-01.

2. Morphological characterization of phage using transmission electron microscopy (TEM) should be done.

• The TEM micrograph (S2 Fig) as well as a description of the overall morphology of the virus has been included in the Bacteriophage Isolation, Host Range Analysis and Morphology Characterization section (lines 178-181).

3. Bacterial reduction assay should be carried out.

• The Bacterial Reduction assay was performed and included in the Bacteriolytic Activity section (Fig 3; lines 196-203) following the reviewer’s recommendation.

4. How did you perform purification of phage particles?

• Our phage purification process involved serial dilution of filtrates exhibiting lytic activity against E. cloacae. Specifically, 100 μL of the phage dilution was mixed with an overnight culture of the host bacterium in 3 mL of soft agar (TSB with 0.4% agar), poured onto TSA plates, and incubated at 37 °C for 24 h. The phage plates, selected based on size and clarity, were transferred to microtubes containing 1 mL of nanopure water and this process was repeated at least three times to confirm that the isolated phages were descendants of a single virion.

This has been included in the Materials and Methos section.

5. Have you identified the structural proteins of isolated phage by SDS-PAGE gel electrophoresis? It will be great to have the proteomic analysis of this phage.

• Following the reviewer’s suggestion we visualized the main protein components of the isolated phage. The image has been included in the new version (S4 Fig) as well as a description of the results (lanes 207-216).

6. Please state the GenBank accession no. of phage genome.

• The GenBank accession no. was declared in the Data availability statement section.

7. What is the percentage of a G + C content of phage genomic DNA.

• The GC content was declared in the Comparative Genomics and Phylogenetic Analysis section.

8. Please describe distinguished genes found in the genome of phage vB_EcRAM-01.

• Additional information on the main traits found in the genome were added in the discussion section.

9. Correct the list of references. Some citations are not present in the reference list.

• References have been corrected in the revised version.

Reviewer #2: 

This manuscript provides data for the biological characterization and genomic analysis of Enterobacter cloacae phage vB_EcRAM-01. Results presented in this study contribute to a greater understanding of the biology of phages and expand our knowledge of Enterobacter cloacae phages. However, the overall format of the article is not consistent with the requirements of the official website of PLOS ONE, please download the manuscript formatting guidelines from the official website of PLOS ONE and modify the article format according to the guidelines.

Major comments

1. Line 19- Enterobacter cloacae should not be abbreviated when it first appears in the keyword section.

• This correction has been made in the current version.

2. Line 39- The font of Enterobacteriaceae should be italic, please check the full text.

• Enterobacteriaceae was italicized.

3. Line 108-110- The experimental conditions in the text are 25 ℃, 50 ℃, 70 ℃ and 90 ℃, but Figure 2B shows 25 ℃, 37 ℃, 50 ℃ and 70 ℃. The temperature unit is not written in Figure 2B. The range of pH in Figure 2C is also inconsistent with the text.

• The correction was made.

4. Figure 2- The unit of the horizontal coordinate of Figure 2 B should be Time (min).

• This has been corrected in the current version.

5. Figure 2- In Figure 2 B, why does the number of phage vB_EcRAM-01 gradually increase at 37 °C?

• Reviewer is right. There was an error when calculating UFP per milliliter when undoing the 10-fold dilutions. This has been fixed in the current version.

6. Line 180- Please write down the correct unit of temperature.

• The correction was made.

7. Line 187-215, 257-266- The description of the genome of phage vB_EcRAM-01 is relatively simple. Please provide more details about the genome.

• Additional information on the main traits found in the genome were added in the discussion section.

8. Figure 3- Please add colors to phage EBPL and phage Margaery to better compare phage vB_EcRAM-01 with these two phages.

• Colors were added to the figure.

9. Figure 4, Figure 5- Please change the font of the genus names in italics on Figures 4 and 5.

• Genus names were italicized.

---

## [Decision Letter · Decision Letter 1]

25 Apr 2024

PONE-D-23-39735R1Characterization of Enterobacter phage vB_EcRAM-01, a new Pseudotevenvirus against Enterobacter cloacae, isolated in an urban river in Panama.PLOS ONE

Dear Dr. Matínez-Torres,

Thank you for submitting your manuscript to PLOS ONE. After careful consideration, we feel that it has merit but does not fully meet PLOS ONE’s publication criteria as it currently stands. Therefore, we invite you to submit a revised version of the manuscript that addresses the points raised during the review process.

**Most of the comments previously raised by both referees have been efficiently addressed in the revised manuscript. Nonetheless, there have been some additional comments by the reviewers which the authors are kindly requested to also take into account. **

We look forward to receiving your revised manuscript.

Kind regards,

Alexandra Lianou, M.Sc., Ph.D.

Academic Editor

PLOS ONE

Journal Requirements:

Reviewers' comments:

Reviewer's Responses to Questions

**Comments to the Author**

1. If the authors have adequately addressed your comments raised in a previous round of review and you feel that this manuscript is now acceptable for publication, you may indicate that here to bypass the “Comments to the Author” section, enter your conflict of interest statement in the “Confidential to Editor” section, and submit your "Accept" recommendation.

Reviewer #1: All comments have been addressed

Reviewer #2: All comments have been addressed

2. Is the manuscript technically sound, and do the data support the conclusions?

Reviewer #1: Yes

Reviewer #2: Yes

3. Has the statistical analysis been performed appropriately and rigorously? 

Reviewer #1: Yes

Reviewer #2: Yes

4. Have the authors made all data underlying the findings in their manuscript fully available?

Reviewer #1: Yes

Reviewer #2: Yes

5. Is the manuscript presented in an intelligible fashion and written in standard English?

Reviewer #1: Yes

Reviewer #2: Yes

6. Review Comments to the Author

**Reviewer #1**: 1. In introduction section; the authors do not go into E. cloacae complex-infecting phages quite enough for this paper. Write more reports about phages specific for E. cloacae complex.

2. Have you done bacterial reduction assay at different MOI?

3. In Methods: state the procedure of phage concentration.

4. In Methods: Please add SDS-PAGE analysis protocol.

5. In result section: you did not write about phage isolation.

6. S2 Fig and S3 Fig are not clear. Please provide clear images with correct resolution.

**Reviewer #2**: Please revise the representation of the values of the ordinate in Fig.2.

And check carefully through out the manuscript.

7. PLOS authors have the option to publish the peer review history of their article (what does this mean?). If published, this will include your full peer review and any attached files.

Reviewer #1: No

Reviewer #2: No

---

## [Author Response · Author response to Decision Letter 1]

31 May 2024

Response to the reviewers:

We appreciate your valuable feedback on our manuscript entitled Characterization of Enterobacter phage vB_EcRAM-01, a new Pseudotevenvirus against Enterobacter cloacae, isolated in an urban river in Panama. We are grateful for the opportunity to address your comments.

Reviewer #1: 1. In introduction section; the authors do not go into E. cloacae complex-infecting phages quite enough for this paper. Write more reports about phages specific for E. cloacae complex.

We thank the reviewer for pointing out this missing piece of information. A full paragraph has been included mentioning recent studies about phages specific for E. cloacae complex in the current version (lines 56-65)

2. Have you done bacterial reduction assay at different MOI?

We acknowledge the importance of investigating the impact of varying MOIs on phage-mediated bacterial reduction. Here we present an exploratory experiment and we have already begun planning subsequent experiments to analyze the dose-response relationship between MOI and bacterial clearance, particularly in combination with different antibiotics that we intend to address in a future paper.

3. In Methods: state the procedure of phage concentration.

The reviewer is right. A section describing the phage concentration procedure has been included in the Methods section in this new version (lines 94-101)

.

4. In Methods: Please add SDS-PAGE analysis protocol.

A section describing the SDS-PAGE analysis protocol has been included in the Methods section in this new version (lines 119-124).

5. In result section: you did not write about phage isolation.

Thank you for pointing this out. A paragraph describing phage isolation has been included in the results section (lines 184-188).

.

6. S2 Fig and S3 Fig are not clear. Please provide clear images with correct resolution.

We acknowledge that the resolution of the TEM image may not meet modern standards due to the limitations of the equipment used. However, we would like to emphasize that nowadays, the classification and identification of bacteriophages are increasingly reliant on genomic analysis rather than on morphological characteristics. Recent studies have demonstrated the importance of complete genome sequencing in the accurate classification of bacteriophages within specific genera. This approach has been widely adopted in the field and has led to a more precise and systematic classification of bacteriophages, overcoming the limitations associated with morphological variability and resolution in TEM images.

Regarding S3 Fig, we agree with the reviewer that the protein bands are quite faint. We know that the precipitation method used in this study doesn’t allow to obtain a more concentrated phage sample. Unfortunately, we do not have access to an ultracentrifuge in Panama that would allow us to concentrate the viruses in order to get a better SDS-PAGE sample.

Reviewer #2: Please revise the representation of the values of the ordinate in Fig.2. And check carefully through out the manuscript.

The values of the ordinates in Figure 2 have been modified.

Once again, we appreciate your constructive feedback, which has strengthened the scientific rigor of our work. We are committed to addressing all concerns raised by the reviewers and ensuring the clarity and completeness of our manuscript.

---

## [Decision Letter · Decision Letter 2]

21 Jun 2024

PONE-D-23-39735R2Characterization of Enterobacter phage vB_EcRAM-01, a new Pseudotevenvirus against Enterobacter cloacae, isolated in an urban river in Panama.PLOS ONE

Dear Dr. Matínez-Torres,

Thank you for submitting your manuscript to PLOS ONE. After careful consideration, we feel that it has merit but does not fully meet PLOS ONE’s publication criteria as it currently stands. Therefore, we invite you to submit a revised version of the manuscript that addresses the points raised during the review process.

We look forward to receiving your revised manuscript.

Kind regards,

Alexandra Lianou, M.Sc., Ph.D.

Academic Editor

PLOS ONE

Journal Requirements:

Additional Editor Comments:

The authors are kindly requested to address the last few minor comments raised by Reviewer #1.

Reviewers' comments:

Reviewer's Responses to Questions

**Comments to the Author**

1. If the authors have adequately addressed your comments raised in a previous round of review and you feel that this manuscript is now acceptable for publication, you may indicate that here to bypass the “Comments to the Author” section, enter your conflict of interest statement in the “Confidential to Editor” section, and submit your "Accept" recommendation.

Reviewer #1: All comments have been addressed

Reviewer #2: All comments have been addressed

2. Is the manuscript technically sound, and do the data support the conclusions?

Reviewer #1: Yes

Reviewer #2: Yes

3. Has the statistical analysis been performed appropriately and rigorously? 

Reviewer #1: Yes

Reviewer #2: Yes

4. Have the authors made all data underlying the findings in their manuscript fully available?

Reviewer #1: Yes

Reviewer #2: Yes

5. Is the manuscript presented in an intelligible fashion and written in standard English?

Reviewer #1: Yes

Reviewer #2: Yes

6. Review Comments to the Author

Reviewer #1: 1. Add the genomic organization of Enterobacter phage vB_EcRAM-01 in the Figure 5

2. How much microliters of concentrated phage solution were used in SDS-PAGE

3. Correct the plaque size in S1 Fig.

Reviewer #2: The manuscript"Characterization of Enterobacter phage vB_EcRAM-01, a new Pseudotevenvirus against Enterobacter cloacae , isolated in an urban river in Panama." revealed a new phage, which should be a potential phage agent. Hope you can do more researches about it.

7. PLOS authors have the option to publish the peer review history of their article (what does this mean?). If published, this will include your full peer review and any attached files.

Reviewer #1: No

Reviewer #2: No

---

## [Author Response · Author response to Decision Letter 2]

3 Jul 2024

Panama, Panama, July 3, 2024

Response to the Editor and reviewers:

Please find below our response to the reviewers.

Reviewer #1: 

1. Add the genomic organization of Enterobacter phage vB_EcRAM-01 in the Figure 5. 

Thank you for your valuable feedback. The genome organization you requested to be added in Figure 5 is already shown in Figure 3, where it is compared with the phage genomes showing higher homology, and the locations of the major genes are indicated. We hope this addresses your concern.

2. How much microliters of concentrated phage solution were used in SDS-PAGE

Ten microliters of the concentrated phage sample were used.

3. Correct the plaque size in S1 Fig.

Plaque size has been corrected in the new version of S1 Fig.

Reviewer #2: The manuscript"Characterization of Enterobacter phage vB_EcRAM-01, a new Pseudotevenvirus against Enterobacter cloacae , isolated in an urban river in Panama." revealed a new phage, which should be a potential phage agent. Hope you can do more researches about it.

Thank you. We plan to isolated new phages against the Enterobacter cloacae complex and use them as cocktails in combination with antibiotics.

---

## [Decision Letter · Decision Letter 3]

23 Jul 2024

PONE-D-23-39735R3Characterization of Enterobacter phage vB_EcRAM-01, a new Pseudotevenvirus against Enterobacter cloacae, isolated in an urban river in Panama.PLOS ONE

Dear Dr. Matínez-Torres,

Thank you for submitting your manuscript to PLOS ONE. After careful consideration, we feel that it has merit but does not fully meet PLOS ONE’s publication criteria as it currently stands. Therefore, we invite you to submit a revised version of the manuscript that addresses the points raised during the review process.

**ACADEMIC EDITOR:**To my view, beyond the two additional comments that were raised by Reviewer #1 (and which the authors are kindly requested to also address), all previously raised comments have been sufficiently and adequately addressed, and there are no further comments raised.

We look forward to receiving your revised manuscript.

Kind regards,

Alexandra Lianou, M.Sc., Ph.D.

Academic Editor

PLOS ONE

Journal Requirements:

Reviewers' comments:

Reviewer's Responses to Questions

**Comments to the Author**

1. If the authors have adequately addressed your comments raised in a previous round of review and you feel that this manuscript is now acceptable for publication, you may indicate that here to bypass the “Comments to the Author” section, enter your conflict of interest statement in the “Confidential to Editor” section, and submit your "Accept" recommendation.

Reviewer #1: All comments have been addressed

2. Is the manuscript technically sound, and do the data support the conclusions?

Reviewer #1: Yes

3. Has the statistical analysis been performed appropriately and rigorously? 

Reviewer #1: Yes

4. Have the authors made all data underlying the findings in their manuscript fully available?

Reviewer #1: Yes

5. Is the manuscript presented in an intelligible fashion and written in standard English?

Reviewer #1: Yes

6. Review Comments to the Author

Reviewer #1: 1. Please add the genomic organization of phage vB_EcRAM-01 with putative open reading frames (ORFs) in a separate figure.

2. Please create a separate table with the predicted functions of ORFs derived from the phage genomic sequence.

7. PLOS authors have the option to publish the peer review history of their article (what does this mean?). If published, this will include your full peer review and any attached files.

Reviewer #1: No

---

## [Author Response · Author response to Decision Letter 3]

1 Aug 2024

1. Add the genomic organization of Enterobacter phage vB_EcRAM-01 in the Figure 5. 

The genome organization you requested has been added as Figure 4 and the rest of the figures have been renamed accordingly. Also, a separate table (sup. Table 1) with the predicted functions of ORFs derived from the phage genomic sequence has been included.

---

## [Decision Letter · Decision Letter 4]

13 Aug 2024

PONE-D-23-39735R4Characterization of Enterobacter phage vB_EcRAM-01, a new Pseudotevenvirus

against Enterobacter cloacae, isolated in an urban river in Panama.PLOS ONE

Dear Dr. Matínez-Torres,

Thank you for submitting your manuscript to PLOS ONE. After careful consideration, we feel that it has merit but does not fully meet PLOS ONE’s publication criteria as it currently stands. Therefore, we invite you to submit a revised version of the manuscript that addresses the points raised during the review process.

We look forward to receiving your revised manuscript.

Kind regards,

Alexandra Lianou, M.Sc., Ph.D.

Academic Editor

PLOS ONE

Journal Requirements:

Reviewers' comments:

Reviewer's Responses to Questions

**Comments to the Author**

1. If the authors have adequately addressed your comments raised in a previous round of review and you feel that this manuscript is now acceptable for publication, you may indicate that here to bypass the “Comments to the Author” section, enter your conflict of interest statement in the “Confidential to Editor” section, and submit your "Accept" recommendation.

Reviewer #1: All comments have been addressed

2. Is the manuscript technically sound, and do the data support the conclusions?

Reviewer #1: Yes

3. Has the statistical analysis been performed appropriately and rigorously? 

Reviewer #1: Yes

4. Have the authors made all data underlying the findings in their manuscript fully available?

Reviewer #1: Yes

5. Is the manuscript presented in an intelligible fashion and written in standard English?

Reviewer #1: Yes

6. Review Comments to the Author

Reviewer #1: 1.In Fig .5, where is the name of the isolated phage? Please add the name of phage vB_EcRAM-01.

2.Please arrange the Figures numbers according to the text of manuscript.

3.In S1 Table, add the numbers of ORFs and check the total numbers of it, add GenBank accession No. and start and stop of each ORF by nucleotide (nt).

7. PLOS authors have the option to publish the peer review history of their article (what does this mean?). If published, this will include your full peer review and any attached files.

Reviewer #1: No

---

## [Author Response · Author response to Decision Letter 4]

19 Aug 2024

In Fig .5, where is the name of the isolated phage? Please add the name of phage vB_EcRAM-01.

The name of phage vB_EcRAM-01 has been now highlighted in red for clarity instead of black.

2.Please arrange the Figures numbers according to the text of manuscript.

We thank the reviewer for noticing that. Figure numbers in the legends have been corrected in this new version of the manuscript

3.In S1 Table, add the numbers of ORFs and check the total numbers of it, add GenBank accession No. and start and stop of each ORF by nucleotide (nt).

Numbers of the ORFs as well as the GenBank accession numbers and start and stop codons for each of the ORFs are now included in S1 table

---

## [Decision Letter · Decision Letter 5]

8 Sep 2024

Characterization of Enterobacter phage vB_EcRAM-01, a new Pseudotevenvirus against Enterobacter cloacae, isolated in an urban river in Panama.

PONE-D-23-39735R5

Dear Dr. Matínez-Torres,

We’re pleased to inform you that your manuscript has been judged scientifically suitable for publication and will be formally accepted for publication once it meets all outstanding technical requirements.

Kind regards,

Alexandra Lianou, M.Sc., Ph.D.

Academic Editor

PLOS ONE

Additional Editor Comments (optional):

Reviewers' comments:

Reviewer's Responses to Questions

**Comments to the Author**

1. If the authors have adequately addressed your comments raised in a previous round of review and you feel that this manuscript is now acceptable for publication, you may indicate that here to bypass the “Comments to the Author” section, enter your conflict of interest statement in the “Confidential to Editor” section, and submit your "Accept" recommendation.

Reviewer #1: All comments have been addressed

2. Is the manuscript technically sound, and do the data support the conclusions?

Reviewer #1: Yes

3. Has the statistical analysis been performed appropriately and rigorously? 

Reviewer #1: Yes

4. Have the authors made all data underlying the findings in their manuscript fully available?

Reviewer #1: Yes

5. Is the manuscript presented in an intelligible fashion and written in standard English?

Reviewer #1: Yes

6. Review Comments to the Author

Reviewer #1: 1. In Fig .5, where is the name of the isolated phage? Please add the name of phage vB_EcRAM-01.

2. In S1 Table, Please revise the start and stop of each ORF by nucleotide (nt).

7. PLOS authors have the option to publish the peer review history of their article (what does this mean?). If published, this will include your full peer review and any attached files.

Reviewer #1: No

---

## [Editor Report · Acceptance letter]

22 Oct 2024

PONE-D-23-39735R5 

PLOS ONE

Dear Dr. Matínez-Torres, 

I'm pleased to inform you that your manuscript has been deemed suitable for publication in PLOS ONE. Congratulations! Your manuscript is now being handed over to our production team.

Kind regards, 

on behalf of

Dr. Alexandra Lianou 

Academic Editor

PLOS ONE